# Anti-Adherence and Antimicrobial Activities of Silver Nanoparticles against Serotypes *C* and *K* of *Streptococcus mutans* on Orthodontic Appliances

**DOI:** 10.3390/medicina58070877

**Published:** 2022-06-30

**Authors:** Rosa Amalia Nafarrate-Valdez, Rita Elizabeth Martínez-Martínez, Erasto Armando Zaragoza-Contreras, José Luis Áyala-Herrera, Rubén Abraham Domínguez-Pérez, Simón Yobanny Reyes-López, Alejandro Donohue-Cornejo, Juan Carlos Cuevas-González, Juan Pablo Loyola-Rodríguez, León Francisco Espinosa-Cristóbal

**Affiliations:** 1Speciality Program in Orthodontics, Department of Dentistry, Biomedical Science Institute, Autonomous University of Ciudad Juarez (UACJ), Envolvente del PRONAF and Estocolmo Avenues, Juarez City 32310, Mexico; rosanafa_@hotmail.com; 2Master Program in Advanced Dentistry, Faculty of Dentistry, Autonomous University of San Luis Potosi, Manuel Nava Avenue, Universitary Campus, San Luis Potosí 78290, Mexico; ritae_martinez@hotmail.com; 3Department of Engineering and Materials Chemistry, Centro de Investigación en Materiales Avanzados, S. C., Miguel de Cervantes 120, Complejo Industrial Chihuahua, Chihuahua 31136, Mexico; armando.zaragoza@cimav.edu.mx; 4School of Dentistry, Universidad De La Salle Bajío, Universidad Avenue, Lomas del Campestre, Guanajuato 37150, Mexico; dr.ayala02@gmail.com; 5Laboratory of Multidisciplinary Dental Research, Faculty of Medicine, Autonomous University of Queretaro, Clavel Street, Prados de La Capilla, Santiago de Querétaro 76176, Mexico; dominguez.ra@uaq.mx; 6Institute of Biomedical Sciences, Autonomous University of Juarez City (UACJ), Envolvente del PRONAF and Estocolmo s/n, Ciudad Juárez 32310, Mexico; simon.reyes@uacj.mx; 7Master Program in Dental Sciences, Stomatology Department, Institute of Biomedical Sciences, Autonomous University of Juarez City (UACJ), Envolvente del PRONAF and Estocolmo s/n, Ciudad Juárez 32310, Mexico; adonohue@uacj.mx (A.D.-C.); juan.cuevas@uacj.mx (J.C.C.-G.); 8Faculty of Dentistry, Autonomous University of Sinaloa, Josefa Ortiz de Domínguez, Ciudad Universitaria, Universitaria, Culiacán 80013, Mexico; juanpablo.loyola8@gmail.com

**Keywords:** silver nanoparticles, *S. mutans* serotypes, dental caries, orthodontics, anti-adherence effect, antimicrobial activity

## Abstract

*Background and Objectives*: *Streptococcus mutans* (*S. mutans*) is the main microorganism associated with the presence of dental caries and specific serotypes of this bacteria have been related to several systemic diseases limiting general health. In orthodontics, white spot lesions (WSL), represent a great challenge for clinicians due to the great fluctuation of their prevalence and incidence during conventional orthodontic treatments. Although silver nanoparticles (AgNP) have been demonstrated to have great antimicrobial properties in several microorganisms, including *S. mutans* bacteria, there is no available information about anti adherence and antimicrobial properties of AgNP exposed to two of the most relevant serotypes of *S. mutans* adhered on orthodontic materials used for conventional therapeutics. The objective of this study was to determine anti-adherence and antimicrobial levels of AgNP against serotypes *c* and *k* of *S. mutans* on conventional orthodontic appliances. *Materials and Methods*: An AgNP solution was prepared and characterized using dispersion light scattering (DLS) and transmission electron microscopy (TEM). Antimicrobial and anti-adherence activities of AgNP were determined using minimal inhibitory concentrations (MIC) and bacterial adherence testing against serotypes *c* and *k* of *S. mutans* clinically isolated and confirmed by PCR assay. *Results*: The prepared AgNP had spherical shapes with a good size distribution (29.3 ± 0.7 nm) with negative and well-defined electrical charges (−36.5 ± 5.7 mV). AgNP had good bacterial growth (55.7 ± 19.3 µg/mL for serotype *c*, and 111.4 ± 38.6 µg/mL for serotype *k*) and adherence inhibitions for all bacterial strains and orthodontic wires (*p* < 0.05). The serotype *k* showed statistically the highest microbial adherence (*p* < 0.05). The SS wires promoted more bacterial adhesion (149.0 ± 253.6 UFC/mL × 10^4^) than CuNiTi (3.3 ± 6.0 UFC/mL × 10^4^) and NiTi (101.1 ± 108.5 UFC/mL × 10^4^) arches. SEM analysis suggests CuNiTi wires demonstrated better topographical conditions for bacterial adherence while AFM evaluation determined cell wall irregularities in bacterial cells exposed to AgNP. *Conclusions*: This study suggests the widespread use of AgNP as a potential anti-adherent and antimicrobial agent for the prevention of WSL during conventional orthodontic therapies and, collaterally, other systemic diseases.

## 1. Introduction

In recent decades the use of fixed orthodontic therapies has increased in some countries and limited information has been specified to determine the risk of potential oral diseases associated with pathogenic microbiota during these dental treatments adhered to these orthodontic appliances [1]. Oral biofilms associated with fixed orthodontic therapies have reported the potential risk factors for health complications identifying the presence of pathogenic species from bacteria, protists, and fungi [1,2]. Some investigations have determined the relationship between the presence of dental caries and periodontal disease during orthodontic treatments [1,3]. Investigators have summarized the clinical and microbiological differences between young subjects with and without fixed orthodontic appliances determining the increased periopathogenic (*P. gingivalis,*
*Prevotella intermedia*, *Tannerella forsythia*, *A. actinomycetemcomitans, Pseudomonas* spp., *P. synxantha, Burkholderia species*, and *Veillonella parvula*) [4] and cariogenic (*Candida* spp, *S. mutans*, *S. sobrinus,* and *Lactobacilli*) [5] bacteria in subjects with fixed appliances [6,7,8]. These changes in the microbial plaque of orthodontic therapies occur, in some cases, immediately after their placement (one week) and consequently increase, gradually, the consistency of microbial colonization three months after the start of treatment [6].

Dental caries is defined as a dynamic and complex process of demineralization and remineralization derived from microbial residual product (lactic acid) deposited on the tooth surface, resulting in overtime, mineral loss, and, possibly, the formation of the cavity [9,10]. Nowadays, dental caries is still considered the most common oral infectious disease affecting several human populations worldwide [10,11]. The complexity of the disease is due to the multiple factors that are associated with the evolution of a bacterial population that goes from a healthy biofilm to a pathological one [12,13]. It is very well known that the primary etiologic agents for dental caries are the streptococci group, particularly *Streptococcus mutans* (*S. mutans*) and *Streptococcus sobrinus* (*S. sobrinus*); but also, other species (*Streptococcus sanguinis*, *Streptococcus gordonii*, and *Streptococcus oralis*) are secondarily implicated to facilitate the development of this disease [14,15]. The bacteria of *S. mutans* is a Gram-positive, chain-arranged, non-mobile, catalase-negative, and rapid producer of compounds derived from carbohydrate metabolism with the ability to change a medium from pH 7 to pH 4.2 in approximately 24 h acting as a fermenter of several fermentable compounds (glucose, lactose, raffinose, mannitol, insulin, and salicin) producing acids (lactic, propionic, acetic, and formic acids) resulting to demineralization process in the tooth enamel [16,17]. One of the most important pathogenic mechanisms of this bacteria is its capacity of adherence using electrostatic and chemical interactions to adhere to dental surfaces and other bacteria from their molecular components (teichoic/lipoteichoic acids, glucosyl/fructosyl transferases, and binding proteins) involved in a complex extracellular matrix [18]. *S. mutans* can be subclassified into various subspecies, which are defined as serotypes and classified into *c*, *e*, *f,* and *k* according to chemical differences in cell wall composition from polysaccharide linkages (rhamnose-glucose) [19]. Serotype *c* is the most predominant type (70–75%) in the human oral cavity, followed by strains *e* (~20%), *f,* and *k* (less than 5%) [20,21,22,23]. *S. mutans* is generally associated with the cause, in specific conditions, bacteremia and infective endocarditis [24,25], but, interestingly, specific *S. mutans* serotypes have been also related to the presence and severity of various systemic diseases. Particularly, type *k* is the last discovered serotype and is characterized by a drastic reduction in the amount of glucose side chains, which have presented a low level of cariogenicity due to alterations in several of the major surface protein antigens [26,27]. Although several studies have determined that this strain can survive in the blood media for a long time resulting in important participation in the pathogenesis of cardiovascular diseases [24,27], renal and hepatic diseases associated with the presence of serotype *k* have been also reported. In addition, studies have reported that the presence of *S. mutans* and specific serotypes, such as type *k,* has been associated with kidney hypertrophy (renal failure), gastrointestinal pathologies (ulcerative colitis) [27], and even serotype *k* could induce aggravated nonalcoholic fatty liver disease [28].

In orthodontics, the demineralization caused by *S. mutans* bacteria is a significant and prevalent problem during conventional orthodontic therapies promoting undesirable effects in dental hard tissues called white spots lesions (WSL) [29] and is still considered one of the most important challenges for clinicians [30]. At the moment of fixing the orthodontic appliances (brackets, wires, elastomeric modules, metallic bands, and others), the oral environment suffers chemical changes prevailing acidogenic bacteria such as *S. mutans* [31,32]. Studies have demonstrated that WSL can appear in incipient stages of the orthodontic treatment and its prevalence is ranged from 2 to 96% [31]. Authors have explained that fixed orthodontic appliances create stagnant areas and make dental cleaning difficult because the irregular surfaces of the brackets, bands, and wires limit the self-cleaning mechanisms of the oral muscles and saliva, promoting the accumulation of bacterial plaque and the colonization of aciduric bacteria, resulting possibly, in active WSLs and, if left untreated, a cavitated lesion can be developed [33]. Several dental treatments have been successfully developed for the control and prevention of dental caries, basically limiting the bacterial growth of *S. mutans* from dental plaque. One of the most important therapies has been the use of topical fluorinated substances in different points of conventional orthodontic treatments [34,35,36,37]. Likewise, gels or varnishes based on sodium fluoride using different concentrations have been effectively used to treat and prevent demineralization of dental structures associated with WSL [34,35,36,37,38,39]; however, the use of these compounds should be interrupted for 15 days before fixed orthodontic appliances [37]. In this sense, fluoride compounds (salt, water, toothpaste, mouthwashes, varnishes, and gels) [34,40,41], antimicrobial agents (chlorhexidine, xylitol) [40,41], and other dental products (toothbrushes, dental floss, and toothpicks) [42] have demonstrated to be excellent dental therapies for the prevention of dental caries; however, the prevalence of this disease remains still high [43]. It is because of that new and novel therapies using different antimicrobial mechanisms to improve the control of *S. mutans* bacteria during conventional orthodontic therapies should be investigated. Silver nanoparticles (AgNP) have been demonstrated to be excellent antimicrobial agents in the biomedical field in a wide variety of microorganisms [44,45], even for oral bacteria [46,47,48]. The action mechanism of AgNP against bacteria has been widely described; however, it remains unclear. Factors such as ion release capacity, biofunctionalization, zeta potential, size, shape, and other physicochemical properties could permit the adhesion of silver ions/particles around of cell wall of bacterial cells causing cell wall disruption and facilitating the silver penetration into the cell attaching to thiols, amino, and hydroxyl groups affecting DNA replication, proteins production, promotion of apoptosis, and, finally, leading cell death [49,50,51,52,53]. In orthodontics, authors have suggested the potential use of AgNP as antimicrobial therapy for the control of *S. mutans* bacteria on surfaces of orthodontic appliances limiting undesirable cariogenic processes during orthodontic procedures [47,54,55]. However, there are no studies that have reported the antimicrobial and anti-adherence activities of AgNP against *S. mutans* bacteria and two of the most important serotypes, such as type *c* and *k*, on surfaces of conventional appliances used for conventional orthodontic treatments. The objective of this work was to evaluate the antimicrobial and anti-adherent activities of AgNP exposed to clinical serotypes *c* and *k* of *S. mutans* bacteria on different types of orthodontic appliances. The obtained results surely will help to propose a better and novel antimicrobial therapy for the control and prevention of the initial stages of dental caries during conventional orthodontic procedures.

## 2. Materials and Methods

### 2.1. Synthesis and Characterization of AgNP

The AgNP sample was prepared using the method previously reported by Espinosa-Cristobal et al. [53]. The preparation starts with a solution of 0.01 M AgNO_3_ (J.T. Baker^TM^ACS, Y06C09, Manufacturer Avantor, Mexico State, Mexico) in a 250 mL beaker. Using magnetic stirring, 10 mL of deionized water with 0.5 g of gallic acid (Sigma-Aldrich, SLCJ6385, Detroit, MI, USA) was added to the initial silver solution. Then, the pH was immediately adjusted to 10 using ammonia hydroxide. The particle size and particle distribution of the obtained solution were determined using dynamic light scattering (DLS) with a HORIBA Scientific’s^TM^ particle analyzer (Nano Partica SZ-100, Franklin Lakes, NJ, USA). The shape of particles was evaluated using a Phillips CM-200 transmission electron microscope (TEM) at an acceleration voltage of 25 kV.

### 2.2. Identification of S. mutans Serotypes by Polymerase Chain Reaction (PCR)

The size of the sample was determined according to the availability and presence of serotypes *c* and *k* using a non-probabilistic and consecutive sampling. Two clinical isolate stocks were provided by the Master’s Degree in Dental Science from San Lui Potosi University, Mexico, with previously informed consent from legally authorized representatives or legal guardians of patients. The study was carried on under relevant guidelines and regulations from Helsinki Declaration. The clinical isolates were collected from saliva samples of children (3–6 years old) with the presence of dental caries (teeth with observable enamel cavitation and no dental fillings). Microbiological samples were grown in *Mitis salivarius* agar and bacitracin (MSB) supplemented with 2% sucrose and 1% potassium tellurite. MSB plates were stored at 37 °C for 48 h and, additionally, for other 48 h at room temperature. Then, the *S. mutans* strain was macroscopically identified using stereoscopic microscopy (Olympus, SD-ILK, Tokyo, Japan). The PCR assay was carried on according to the methods previously reported in 2007 [56] and 2016 [49]. The primers used to identify *S. mutans* and serotypes *c* and *k* were reported by Hoshino et al., 2004 [57] (*S. mutans*, size 433 bp [MKD-F: GGCACCACAACATTGGGAAGCTCAGTT; and MKD-R: GGAATGGCCGCTAAGTCAACAGGAT]); Shibata et al., 2003 [22] (serotype *c*, size 727 bp [SC-F: CGGAGTGCTTTTTACAAGTGCTGG; and SC-R: AACCACGGCCAGCAAACCCTTTAT]); and Nakano et al., 2004 [19] (serotype *k*, size 300 bp [CEFK-F: ATTCCCGCCGTTGGACCATTCC; and K-R: CCAATGTGATTCATCCCATCAC]), respectively. The positive controls used for each DNA serotype were GS5 and NG71 for serotype *c* and FT1 for serotype *k*.

### 2.3. S. mutans Suspensions and Antibacterial Test

Standardized bacterial suspensions of *S. mutans* serotypes *c* and *k* containing ~1.5 × 10^8^ CFU/mL were used for all experimental procedures. Once we identified and classified both *S. mutans* serotypes by PCR, each serotype strain was cultured in Müller–Hinton broth (MH, BD Difco, Manufacturer Dickinson and Company, Sparks, Maryland, USA) for 24 h at 37 °C before the test. The antibacterial activity was obtained through minimal inhibitory concentrations (MIC) according to the previously reported method [53]. In 96-well microdilution plates; 200 µL (1070 µg/mL of AgNP for both Ag samples) were added to the first column and diluted 1:1 with MH medium containing 2% sucrose until the eleventh column (penultimate column). Finally, 100 µL of each *S. mutans* serotype suspension at ~1.5 × 10^6^ CFU/mL was inoculated to all wells and incubated for 24 h at 37 °C. The first (AgNP/bacteria/no MH medium) and twelfth (MH medium/bacteria/no AgNP) columns were used as positive and negative controls, respectively. The MIC value was visually identified and confirmed using a stereoscopic microscope (Motic SMZ-168) considering the last well that presented observable turbidity compared to positive and negative control wells, respectively. All antibacterial tests were carried out in triplicate.

### 2.4. Preparation of Dental Samples

Sixteen freshly extracted molars were collected and stored in saline solution (NaCl 0.85%) to prevent dehydration using a non-probabilistic and consecutive sampling. Molars with evident structure alterations and dental restorations were excluded. Molars were carefully cleaned, sonicated, and stored in saline solution at 4 °C. Crowns of each molar were cut in individual enamel blocks of 7 × 7 mm (49 mm^2^) using a diamond disc with irrigation. Peripheric areas of blocks were standardized and coated with nail polish (two coatings) except for a 6 × 6 mm (36 mm^2^) window without nail polish. Then, the windows of each enamel block were dried with fresh air and 5.25% sodium hypochlorite was gently added to the surface for 60 s rubbing the surface using sterilized rubber swaps; finally, surfaces were washed using distilled water for 30 s and dried with pressured air. A conventional total etching system was carried out using 35% phosphoric acid (Ultra-etch^TM^, Ultradent^TM^) for 15 s on all enamel surfaces, immediately the samples were washed with distilled water for 30 s and dried with pressured air. In addition, an adhesive system for orthodontics (Ortho Solo^TM^, Ormco^TM^) was placed on the enamel surfaces with a micro brush and light-cured for 20 s (LED lamp, intensity of 800 mW/cm^2^). Then, stainless steel orthodontic brackets recently purchased for premolars (Ah-Kim-Pech^TM^, Roth slot 0.018) were bonded to the enamel window using a resin-based composite for orthodontic treatments (Enlight^TM^, Ormco^TM^) compressing lightly to the enamel window with a metallic bracket-holder clamp and light-cured for 30 s. The position of each orthodontic bracket was centered on the enamel window, resulting in free space (dental enamel surface without nail polish) approximately 1 mm around the orthodontic bracket. For each sample, stainless steel (SS, Ah-Kim-Pech^TM^), nickel-titanium (NiTi, Ah-Kim-Pech^TM^), and cupper- nickel-titanium (CuNiTi, Ah-Kim-Pech^TM^) orthodontic wires of 5 mm of length were placed and fixed on the bracket slot using individual elastomeric modules (Ah-Kim-Pech^TM^). Additionally, brackets, elastomeric modules, or orthodontic wires with fractures, bent, twisted, wry, or other evident topographic alterations were also excluded. Finally, all dental samples were placed into three different plastic tubes and sterilized in an autoclave (120 °C/15 min). Thus, the study groups were randomly divided according to the presence and type of orthodontic wire used with four samples per group: (1) SS group (n = 4 samples), (2) NiTi group (n = 4 samples), (3) CuNiTi group (n = 4 samples), and (4) control group with no orthodontic wire (n = 4 samples).

### 2.5. Adherence Testing

In an aseptic environment using a laminar air-flow chamber, each orthodontic sample was individually placed into a plastic tube of 3 mL; then, 1 mL of brain–heart infusion broth (BHI Difco, Manufacturer Dickinson and Company, Sparks, Maryland, USA) and 825 µL of the AgNP solution or 125 µL of distilled water (used as negative control) were added to the tube, respectively. In addition, 100 µL of each standardized suspension for serotype *c* and *k* (1.5 × 10^8^ CFU/mL) were complemented to all tubes and incubated for 24 h at 37 °C. After this time, each orthodontic sample was removed from the tubes and placed into another sterilized plastic tube with 1 mL of phosphate-buffered saline solution (PBS) and sonicated for 5 min using an ultrasonic cleaner (Bransonic 1510R-DTH, 70 W ultrasound, 37 kHz, Degas function, Branson Ultrasonic Corporation, Danbury, Connecticut, USA). Thus, the adhered bacterial cells on the surface of dental enamel, bracket, elastic module, and orthodontic wires were detached and dispersed in the media using a PBS solution. This suspension was diluted 1000 times using serial dilutions with PBS, and 100 µL of each suspension was placed and extended on BHI agar plates and, finally, incubated for 24 h at 37 °C. The adherence activity was determined by colony-forming unit count (CFU/mL) in triplicate for serotype *k* and duplicate for serotype *c*.

### 2.6. SEM and AFM Analysis

To evaluate the topographic characteristics of SS, NiTi, and CuNiTi orthodontic wires, new and dried wires for each type were examined by scanning electron microscopy (SEM, JEOL JSM-5300 LV, Tokyo, Japan) operating at an accelerating voltage of 15 kV using magnifications of ×100, ×200, and ×5000 in magnitudes. Additionally, to explore cell wall alterations in bacterial cells exposed to AgNP, a bacterial sample using serotype *c* of *S. mutans* was prepared and examined by an atomic force microscopy instrument (AFM, Nanosurf Easy Scan 2, SPM Electronics, Liestal, Switzerland) using a basic module, contact mode, static force, and scan head type of 70 µm with a ContAl-G AFM probe model (Budget Sensors, Bulgaria) containing rotated monolithic silicon probe with symmetric tip shape, chip-size of 3.4 × 1.6 × 0.3 mm, an aluminum reflex coating (30 nm thick). All values and images from AFM were obtained by Nanosurf Easy Scan 2 software (Version 3.1.1.22, Nanosurf AG, Gräubernstrasse, Liestal, Switzerland).

### 2.7. Statistical Analysis

All data from antimicrobial and anti-adherence assays were expressed in mean and standard deviations. Normal distribution from variables was determined by the Shapiro–Wilk test. All statistical comparisons between independent groups and subgroups for antimicrobial agents (AgNP and distilled water as control group), orthodontic wires (N/A, SS, CuNiTi, and NiTi) as well as *S. mutans* serotypes (*c* and *k*) were analyzed by Mann–Whitney U test for non-parametric values. The statistical program used was IBM-SPSS software (SPSS^®^, version 25, Chicago, IL, USA). Groups were considered significantly different when *p* < 0.05.

## 3. Results

This section may be divided by subheadings. It should provide a concise and precise description of the experimental results, their interpretation, as well as the experimental conclusions that can be drawn.

### 3.1. Characterization of AgNP

Characteristics of AgNP used in this study are shown in Table 1 and Figure 1. DLS results showed a narrow and single peak indicating uniform sizes and good distribution of particles having an average size of 29.3 ± 0.7 nm. For TEM micrographs, spherical and pseudospherical shapes of particles can be uniformly observed with a negative and well-defined electrical charge (−36.5 ± 5.7 mV). MIC results indicate that AgNP had good antibacterial effects for both serotypes (Table 1). Although the serotype *k* presented more bacterial resistance to AgNP (111.4 ± 38.6 µg/mL) than serotype *c* (55.7 ± 19.3 µg/mL), no statistical differences were identified (*p* = 0.09), indicating similar antimicrobial sensibility for both serotypes to AgNP.

### 3.2. Adherence Testing

Figure 2 and Table 2 show the anti-adherence inhibition activity of AgNP among *S. mutans* bacteria and its serotypes on orthodontic wires. In general, the AgNP sample had statistically good anti-adherence effects compared to the control group (Figure 2a). Moreover, the AgNP demonstrated to have similar adherence inhibition for any orthodontic wire (*p* > 0.05); however, for the control group, some variations in the bacterial adhesion were statistically found (Figure 2b). In that way, the SS wire promoted the highest bacterial adherence (289.3 ± 306.8 CFU/mL) followed by NiTi (202.0 ± 38.76 CFU/mL), dental enamel with no orthodontic wire (164.5 ± 19.25 CFU/mL), and, finally, the orthodontic appliance with the lowest bacterial adhesion was the CuNiTi group (6.5 ± 7.5 CFU/mL) (Figure 2c). Furthermore, the presence of AgNP and the CuNiTi arcs significantly showed the best anti-adherence inhibition activity (Figure 2d).

Figure 3 and Table 2 show adherence inhibition activity of AgNP against serotypes *c* and *k* of *S. mutans* bacteria. Serotype *k* presented lightly more bacterial adhesion (123.9 ± 193.1 CFU/mL) than serotype *c* (44.0 ± 71.4 CFU/mL); therefore, no significant differences were determined (Figure 3a,d). In addition, the presence of AgNP promoted significantly better inhibition activity against serotypes *c* (0.5 ± 0.7 CFU/mL) and *k* (4.2 ± 7.7 CFU/mL) compared to the control group (87.5 ± 80.9 and 243.5 ± 216.1 CFU/mL, respectively) (Figure 3b). Finally, the presence of specific orthodontic wires was also associated with the adhesion capacity of *S. mutans* serotypes (Figure 3c). Although the serotype *k* was almost always more adherent to the surfaces of all orthodontic arcs than serotype *c*, the SS group demonstrated to have, statistically, more adhered bacteria for serotype *k* (293.1 ± 302.6 CFU/mL) compared to serotype *c* (4.8 ± 6.0 CFU/mL), while the CuNiTi arcs showed the lowest number of bacteria adhered for both serotypes (6.6 ± 7.3 and 0 ± 0 CFU/mL, respectively). These results suggest that the inhibition of bacterial adhesion of *S. mutans* serotypes *c* and *k* could be related to the presence of AgNP, specific orthodontic appliances (such as CuNiTi arcs), and specific serotypes of *S. mutans* (Figure 4).

### 3.3. SEM and AFM Analysis

To explore the surface characteristics of the metallic appliances used in this study, Figure 5 shows the topographical evaluation of orthodontic wires using SEM. More topographic irregularities, larger craters, and higher roughness were observed on surfaces of CuNiTi wires (Figure 5d–f) compared to NiTi arcs (Figure 5g–i) in which scarce scratches, small pores and, a roughness much less evident were detected. The SS orthodontic arcs showed the smoothest surfaces with minimum scratches (Figure 5a–c). On the other hand, Figure 6 shows an exploration of the antimicrobial effect of AgNP on serotype *c* of *S. mutans* using AFM analysis. For the control group, well-organized and distributed bacterial cells can be observed finding intact cell membranes with rounded shapes. In addition, strong cell–cell bacterial adhesion can be easily observed for both images of the control group making a well-defined bacterial monolayer, producing even, good sharpness, and high-definition AFM images (Figure 6a,b). Therefore, bacterial cells of *S. mutans* exposed to AgNP showed alterations in cell–cell adherence and several morphological irregularities around the cell walls. Interestingly, these micrographs from bacterial cells exposed to AgNP had significantly more interference lines during the scanning of AFM instruments promoting images with poor sharpness and low image qualities resulting in more blurry images (Figure 6c,d) than those bacterial cells with exposition to AgNP. These AFM images suggest that bacterial cells exposed to AgNP could have weaker cell membranes than non-treated cells interfering in the correct position and microtopographic evaluation of the AFM tip.

## 4. Discussion

This study determined that AgNP can significantly inhibit bacterial growth and promote a good anti-adherence effect against serotypes *c* and *k* of *S. mutans* strain on several orthodontic wires using conventional orthodontic appliances. Although the AgNP had exceptional anti-adherence activity for all types of orthodontic arches and both *S. mutans* serotypes (*c* and *k*), the best anti-adherent level was principally reached by CuNiTi arch and serotype *c* of *S. mutans* bacteria while the poorest activity of AgNP was observed in SS wire and serotype *k*. In addition, topographic irregularities were found in the CuNiTi arch, principally, while morphology and adhesive variations from *S. mutans* bacteria exposed to AgNP were identified. In general, the AgNPs used in this study showed good antimicrobial and anti-adherence properties of serotypes *c* and *k* of the *S. mutans* bacteria, even with the presence of orthodontic devices that could improve the adhesion capacity of this bacterial strain.

Several studies have already demonstrated the excellent antimicrobial activity of AgNP against a wide broad of microorganisms, surfaces, and conditions [49,50,51,52,53]. Therefore, there are scarce reports that have evaluated the effect of AgNP on the adhesion of *S. mutans* bacteria and their serotypes *c* and *k* on the surfaces of brackets, different varieties of archwires, and elastomeric modules used routinely in conventional orthodontic treatments. It is very well known that specific characteristics of AgNP such as well-defined electrical charge, narrow size, good particle distribution, the concentration of precursor, stabilizing agent, elevated pH values, and other conditions can be directly associated with their antimicrobial and anti-adherence properties of this nanomaterial permitting non-agglomerated particles [54,55,58]. Our characterization results indicate that the AgNP sample used had generally a good distribution and uniform sizes and shapes. Zeta potential results suggest that AgNP may have a lower risk of agglomeration due to a well-defined electrical load (−36.5 ± 5.7 mV), under other reported studies that found similar results. We assume that the concentration of AgNO_3_ (0.01 M), elevated pH (pH 10), and the specific concentration of gallic acid as stabilizing agent (0.5 g) included in a 100 mL of deionized water can create Ag nanoparticles with electrical charges in its surface that promote better properties associated with the stability of the particles. On the other hand, studies have also determined that a coating of AgNPs in human dentin and enamel can significantly prevent the formation of bacterial biofilm on the surface of the dentin, as well as inhibit the bacterial growth around it. These authors described that the smaller AgNP may have the ability to release more Ag+ ions with a larger surface area, which increases their antimicrobial effect [59,60,61,62,63]. Particularly in the orthodontic field, scarce studies have evaluated the antimicrobial and anti-adherent effects of AgNP on orthodontic appliances such as brackets, wires, adhesives, elastomeric modules, resin-based composites, and other orthodontic materials. Studies have evaluated the mechanism of joining dental materials with AgNP to prevent the adhesion of specific microorganisms to reduce biofilm formation [64,65]. Some of these works have used resin composites containing filler implanted with silver ions [66] or a composite adhesive containing silica nanofillers and AgNP, all of them, against cariogenic bacteria associated with dental caries in orthodontics [67]. These works suggested that the materials with AgNP had significant reductions in bacterial growth than those with conventional materials [66,67,68]. Another study reported that AgNP inhibited bacterial adhesion and the growth capacity of *S. mutans* bacteria on the surfaces of brackets and different types of orthodontic wires associating the anti-adherence ability and the antimicrobial effect of AgNP with the size of the particle, type of orthodontic appliance with the topographic conditions involved in each wire [69]. A recent work studied the anti-adherent and antibacterial properties from 300 samples of different orthodontic brackets with AgNP against *S. mutans* and *S. sobrinus* bacteria resulting in variations in the anti-adherent and antibacterial properties of AgNP according to specific orthodontic brackets and bacterial strain [45]. These authors concluded that AgNP on the surface of orthodontic brackets can be used to prevent dental plaque formation and consequently dental caries during orthodontic treatments [45]. Additionally, serotypes of *S. mutans* have been widely studied; however, few works have evaluated the relationship between antimicrobial and anti-adherence properties of AgNP on serotypes of *S. mutans*. These serotypes have been classified by microbiological differences, in part on the chemical composition included in the cell wall [19,20]. Authors have reported that the antimicrobial sensitivity of *S. mutans* bacteria and its serotypes is related to the particle size and specific serotypes, concluding that serotypes *e* and *k* were generally the most resistant strains compared to *c* and *f* [49,70]. Our results indicated significant and good adherence inhibition and antimicrobial activities of AgNP samples against *S. mutans* clinical isolates and its serotypes *c* and *k* on orthodontic wires, modules, and bracket surfaces (*p < 005*). In addition, this study found that serotype *k* has greater bacterial adhesion capacity and antimicrobial resistance compared to serotype *c*. In addition, Figure 6 shows basically numerous interference lines in images from serotype *c* of *S. mutans* bacteria exposed to AgNP compared to bacteria with no exposition. This result might be involved with the alteration of *S. mutans* bacteria caused mainly by AgNP exposition, producing alterations in the cell wall structure associated with the poor ability of adhesion and structural weakness to the tip of AFM, dragging and disrupting the cell wall of *S. mutans* strain and, finally, to produce AFM micrographs with many interference lines. Authors have already proposed that AgNP have a greater antimicrobial effect depending on the contact surface. These investigators have concluded that the smaller nanoparticles may have the ability to drastically penetrate the cell membrane, inhibiting the enzymes of the respiratory system chain of bacterial cells [71]. We considered there are several factors that could be involved in the antimicrobial and anti-adherence mechanism of AgNP. In this sense, it is possible that AgNPs could have penetrated the cell membrane of *S. mutans* bacteria, affecting the metabolic system of these bacteria, preventing the production of extracellular polysaccharides (principal bacterial adhesion agents) and specific metabolic processes for the development of binding structures of *S. mutans* in surfaces and other bacteria [47,51,59]. However, other conditions such as sociodemographic, genetic, habits, and other personal characteristics from patients could also be related [46]. In addition, it is possible to suggest that serotype *k* has better microbiological components to facilitate the adhesion basically on surfaces of orthodontic appliances modulating the resistance or sensibility to AgNP.

The control of adherence and bacterial growth of *S. mutans* strain and its serotypes organized in dental biofilms plays an important role in the prevention of WSLs leading to more effective and predictable orthodontic therapies. Orthodontic appliances such as brackets, wires, elastomeric modules, and others, also intervene in the success of treatments. Particularly, some studies have described that specific compounds in the fabrication of orthodontic appliances can be related to the bacterial adherence of cariogenic bacteria associated with their manufacturing and complex design [45]. The SS has been identified as a material with a high potential for microbial adherence due to surface energy [72,73]; but also, it has demonstrated to have a low adherence affinity with *S. mutans* compared to other materials such as plastics and ceramics [74]. In addition, authors have reported that CuNiTi wire had higher levels of bacterial adherence compared to NiTi when they were exposed to the *S. mutans* strain, possibly due to differences in rougher surface and higher surface free energy [75]. Furthermore, the surface roughness, apparent surface free energy (SFE), and bacterial adherence of NiTi and SS wires were also reported, suggesting that NiTi wires showed more increased levels of bacterial adhesion of *S. mutans* and SFE than SS arches [76]. Our results suggest that AgNP demonstrated to have a great ability to inhibit the bacterial adhesion and growth capacity of *S. mutans* bacteria associated with specific *S. mutans* serotype and orthodontic wire. This could suggest a combined antimicrobial mechanism of AgNP, such as adhering to and penetrating the cell membrane of *S. mutans* strain affecting the metabolic system of the bacteria, preventing the production of polysaccharides, limiting the adherence mechanism to survival [59,69], but also at the same time, the metallic copper can synergically generate an antimicrobial activity against the oral bacteria [77]. Although the topographic characteristics of CuNiTi have already been described as having more free energy surfaces and more roughness than SS and NiTi wires [75,78] and, possibly, being able to increase the bacterial adherence [69,76]; authors have also reported the antimicrobial and toxic activity of metallic copper, indicating that bacteria and viruses are effectively killed on the metallic copper surfaces [79,80]. Additionally, this could indicate that other metallic materials included in the orthodontic wires might directly intervene in the antimicrobial and anti-adherence mechanism of AgNPs producing, even, failures in the adhesion of *S. mutans* bacteria and its serotypes.

With these results obtained, we could suggest an effective control of *S. mutans* adhesion in orthodontic appliances through in vitro tests using two of the most important serotypes (*c* and *k*). This study could recommend the potential use of AgNP (through mouthwash or dental varnish applications) to promote better control of bacterial growth and interfere with the adhesion ability of serotypes *c* and *k* of *S. mutans* bacteria and prevent the initial formation of dental caries on conventional orthodontic therapies. However, it is very necessary to include other experimental approaches such as the inclusion of serotypes *e* and *f*, the use of clinical samples (clinical biofilms) with particular sociodemographic and systemic characteristics, other oral infectious diseases, a higher number of samples, more characterization methods to determine different properties of AgNP and others micromolecular assays, or even, the use of nanostructured silver in combination with others organic or inorganic biomaterials such as biomimetic hydroxyapatite for remineralization process in orthodontics [81]. This could generate a better understanding of the action mechanism of AgNP to intervene in the bacterial adhesion and bacterial growth capacity of *S. mutans* serotypes to different types of surfaces conventionally used for regular orthodontic therapies facilitating the control and prevention of dental caries and, probably, the prevention of other systemic diseases associated with the presence of serotypes *c* and *k* during orthodontic treatments.

## 5. Conclusions

This study demonstrated that the synthesized AgNP had well-defined shapes, sizes, and electrical charges in a homogeneous colloidal solution, The AgNP inhibited significantly the bacterial adhesion and growth capacity of serotypes *c* and *k* of *S. mutans* bacteria on surfaces of brackets, elastomeric modules, and archwires used conventionally for orthodontic treatments (SS, NiTi, and CuNiTi), producing anti-adherent and antimicrobial properties of AgNP associated, surely, with the size, shape, and electrical charge of the particle. Moreover, the orthodontic wire with the highest bacterial adherence was presented by SS, followed by NiTi and, the archwire with the lowest adhesion capacity was CuNiTi. The most resistant strain to AgNP was serotype *k*, while the most sensible bacteria was serotype *c*. According to our knowledge, this is the first study that determined the antimicrobial and anti-adherent activities of AgNP against serotypes *c* and *k* of *S. mutans* bacteria adhered to different appliances regularly used for conventional orthodontic therapies for the control of initial dental caries, suggesting particular mechanisms from AgNP related to their physicochemical properties, microbiological particularities of *S. mutans* serotypes, and specific topographic characteristics of orthodontic appliances. Although the AgNP used in this study could be a potential bactericidal and anti-adherence agent for the control and prevention of dental caries during conventional orthodontic procedures and, possibly, to prevent other systemic alterations, different in vitro and in vivo experiments using other simulated environments are highly recommended. The preparation, design, presentation, therapeutic application, and biomedical approaches of new and novel antimicrobial agents using nanostructured materials based on Ag should be the point of future investigations to generate better, safe, and more sustainable therapies in the biomedical field, particularly helping in the control of oral diseases in orthodontics.

## Figures and Tables

**Figure 1 medicina-58-00877-f001:**
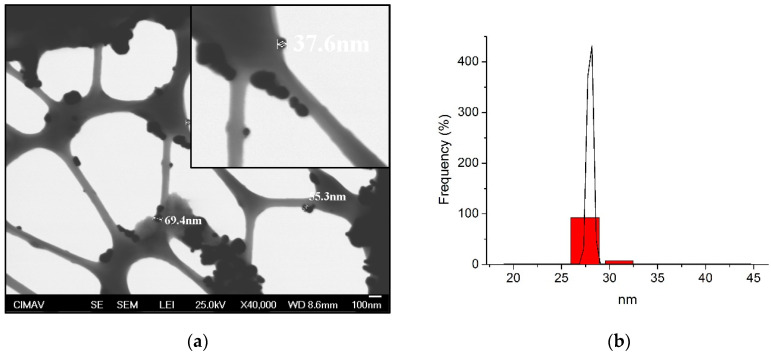
Dynamic light scattering (DLS) and transmission electron microscopy (TEM) analysis of silver nanoparticles (AgNP): (**a**) TEM micrographs; (**b**) size distribution by DLS. Figure 1a shows spherical shapes of AgNP with a well-distribution of particle sizes between ~37.6 up to ~69.4 nm. In Figure 1b, DLS results described a narrow-size distribution of particles identifying a single and slim peak with size averages of 29.3 ± 0.7 nm. TEM and DLS analysis determined that the AgNP has well-organized characteristics with uniform shapes and well-distributed sizes.

**Figure 2 medicina-58-00877-f002:**
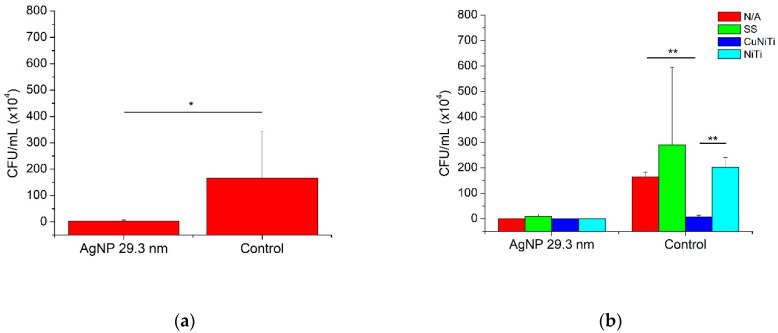
The anti-adherence activity of AgNP according to the orthodontic wire. AgNP = silver nanoparticles; N/A = no archwire; SS = stainless steel; CuNiTi = copper/nickel/titanium; and NiTi = nickel/titanium. Statistical comparisons between groups (AgNP and distilled water) and subgroups (orthodontic wires and *S. mutans* serotypes) were analyzed by Mann–Whitney U test for non-parametric values. Statistical differences were presented using asterisks (* = *p* < 0.05; ** = *p* < 0.01).

**Figure 3 medicina-58-00877-f003:**
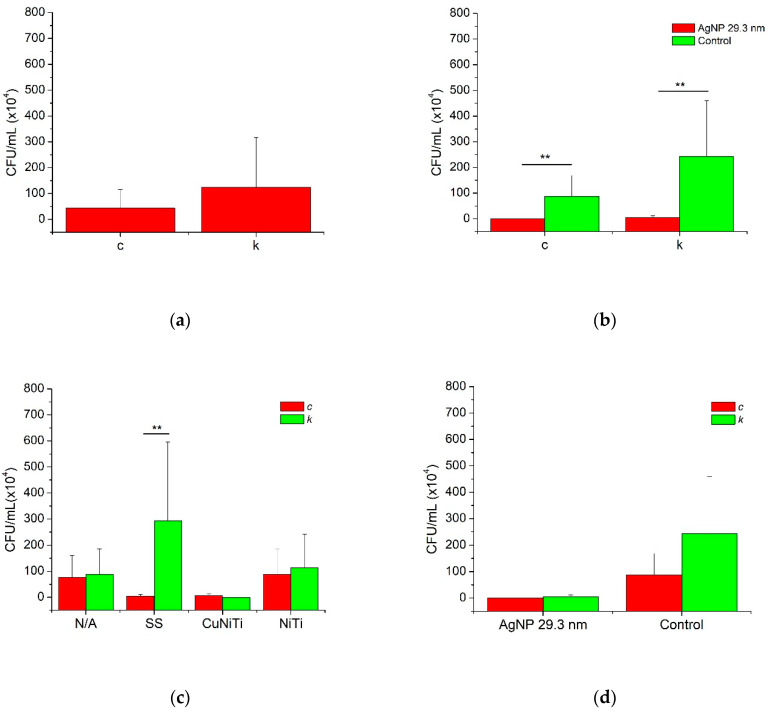
The anti-adherence activity of AgNP according to *S. mutans* serotypes. AgNP = silver nanoparticles; N/A = no archwire; SS = stainless steel; CuNiTi = copper/nickel/titanium; and NiTi= nickel/titanium. Statistical comparisons between groups (AgNP and distilled water) and subgroups (orthodontic wires and *S. mutans* serotypes) were analyzed by Mann–Whitney U test for non-parametric values. Statistical differences were presented using asterisks (** = *p* < 0.01).

**Figure 4 medicina-58-00877-f004:**
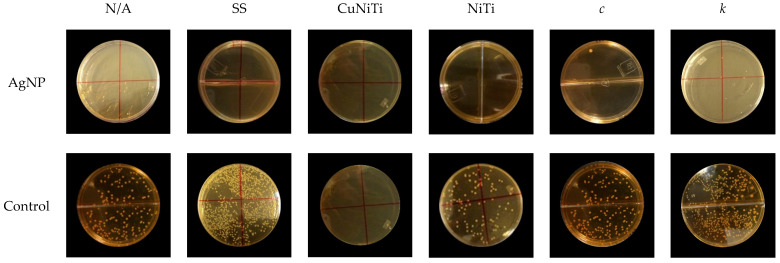
Representative agar plates of the anti-adherence activity of AgNP according to orthodontic wires and *S. mutans* serotypes. AgNP = silver nanoparticles; N/A = no archwire; SS = stainless steel; CuNiTi = copper/nickel/titanium; and NiTi= nickel/titanium.

**Figure 5 medicina-58-00877-f005:**
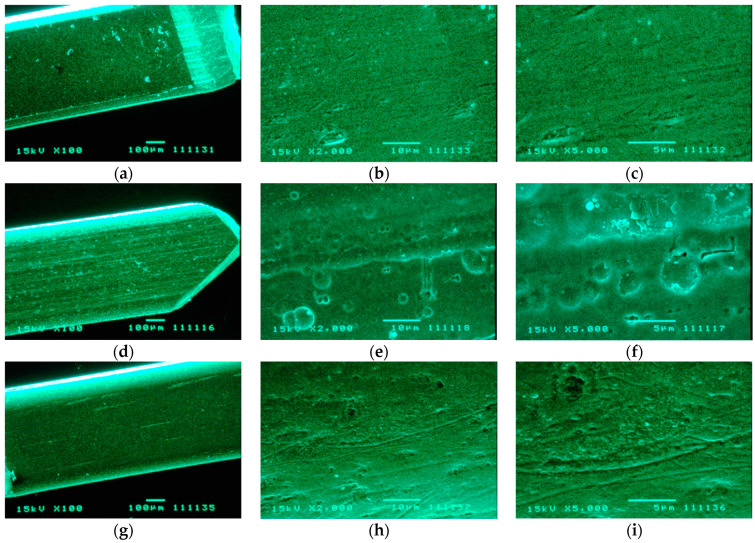
Topographic evaluations of orthodontic appliances by scanning electron microscopy (SEM): (**a**) Stainless steel, ×100; (**b**) stainless steel, ×2000; (**c**) stainless steel, ×5000; (**d**) copper-nickel-titanium, ×100; (**e**) copper-nickel-titanium, ×2000; (**f**) copper-nickel-titanium, ×5000; (**g**) nickel-titanium, ×100; (**h**) nickel-titanium, ×2000; (**i**) nickel-titanium, ×5000. CuNiTi wires (**d**–**f**) presented more irregularities on the surface with the deepest and largest craters identifying these surfaces with the highest roughness. NiTi arcs (**g**–**i**) showed topographic alterations slightly lower than CuNiTi samples finding scarce scratches, smaller pores, and cracks resulting in roughness characteristics more acceptable. SS archwires (**a**–**c**) had scarce irregularities in their topographic resulting in the smoothest surfaces with the lowest values of roughness.

**Figure 6 medicina-58-00877-f006:**
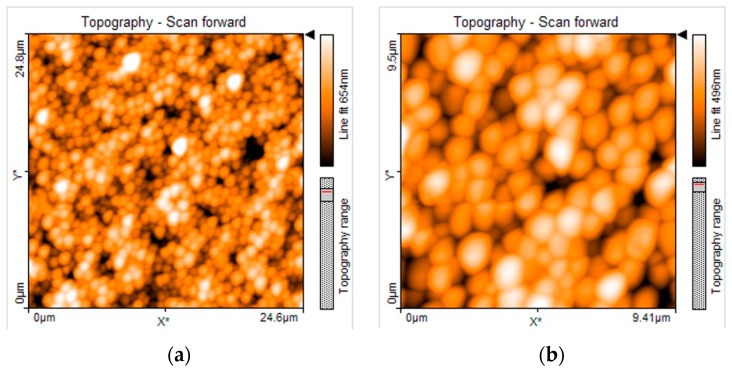
Antimicrobial analysis of serotype *c* of *S. mutans* strain by atomic force microscopy (AFM): (**a**,**b**) Control group; (**c,d**) AgNP 29.3 nm. To explore the action mechanism of AgNP, AFM analysis was carried out on bacterial cells of *S. mutans* exposed to AgNP. For bacterial cells non-treated with AgNP (control group) well-organized cells and non-altered cell morphology were identified (**a**,**b**). On the other hand, bacterial cells treated with AgNP (**c**,**d**) demonstrated to have more observable alterations in cell–cell adherence and more morphological alterations compared to non-treated bacterial cells. Even, bacterial cells exposed to AgNP could have had modifications to their cell membranes weakening the inner structure of their cell walls and producing relevant interferences in the position and scanning from the AFM tip, obtaining AFM images with considerable interference lines, and low evaluation quality.

**Table 1 medicina-58-00877-t001:** Characteristics of AgNP used in the study.

AgNP (nm)	DLS Diameter (nm)	Shape	Initial Concentration (µg/mL)	Zeta Potential (mV)	MIC in Serotype *c* (µg/mL)	MIC in Serotype *k* (µg/mL)
29.3	29.3 ± 0.7	Spherical	1070	−36.5 ± 5.7	55.7 ± 19.3	111.4 ± 38.6

Dynamic light scattering is expressed in mean and standard deviation. Zeta potential is expressed in mean and zeta deviation.

**Table 2 medicina-58-00877-t002:** The anti-Adherence activity of AgNP on *S. mutans* serotypes.

	AgNP 29.3 nm (CFU/mL × 10^4^)	Control (CFU/mL × 10^4^)	Total (CFU/mL × 10^4^)
Serotype			
*c*	0.5 ± 0.7 *	87.5 ± 80.9	44.0 ± 71.4
*k*	4.2 ± 7.7 *	243.5 ± 216.1	123.9 ± 193.1
Archwire			
N/A	0.3 ± 0.8 *	164.5 ± 19.2 ^a^	82.4 ± 86.7
SS	8.6 ± 9.1	289.3 ± 306.8	149.0 ± 253.6 ^a^
CuNiTi	0.1 ± 0.4	6.5 ± 7.5 ^a,b^	3.3 ± 6.0 ^a^
NiTi	0.3 ± 0.8 *	202.0 ± 38.7 ^b^	101.1 ± 108.5
Total	2.3 ± 5.6 *	165.5 ± 178.4	-

All data are expressed in mean and standard deviation. AgNP = silver nanoparticles; N/A = no archwire; SS = stainless steel; CuNiTi = copper/nickel/titanium; and NiTi = nickel/titanium. Statistical comparisons between groups (AgNP and distilled water) and subgroups (orthodontic wires and *S. mutans* serotypes) were analyzed by Mann–Whitney U test for non-parametric values. For rows, one asterisk indicates a statistical difference with the control group (*p* < 0.05). For columns, similar lowercase letters indicate statistical differences (*p* < 0.05). - not applicable.

## Data Availability

All data obtained from this study can be found in the research archives of the Master’s Program in Dental Sciences of the Autonomous University of Ciudad Juarez and can be requested through the corresponding author.

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
