# Peer review of "Anti-Adherence and Antimicrobial Activities of Silver Nanoparticles against Serotypes C and K of Streptococcus mutans on Orthodontic Appliances"

_medicina, 2022, doi:10.3390/medicina58070877_

Round 1

Reviewer 1 Report

Manuscript of considerable interest for the dental sector, especially for orthodontists and dental hygienists.

Before you can actually evaluate the possibility of a publication you need a peer review.

Abstract: we need the data obtained in a clearer way, also poisoning the numbers.

Keywords: they are few, add specific ones

Introduction: it is necessary to add how the microbiota of the oral cavity changes in the orthodontic patient, both in the temporal range and on the basis of age, it is necessary to analyze all the remineralizing and antibacterial systems used at the moment, specifying the interruption of any fluorinated substance 15 days before the bonding (see Cossellu et al.)

Materials and methods: well described

Results: to make the results obtained more visible, perhaps by adding a summary table

Discussion: Add as future goals, the use of silver in combination with biomimetic hydroxyapatite as already studied by Prof Scribante et al.

Conclusions: to emphasize the results obtained and to be a little more avant-garde

Bibliography: add the required references

Reviewer 2 Report

Dear Authors, 

The study is interesting to read, however, there are a few concerns. 

Abstract section-

1. Elaborate on the "characterized" terminology used in the material & Method section of the abstract. 

2. Do not merge the results of different aspects in one sentence. For example, "serotype k and SS wires demonstrated".... mention in different sentences. 

3. Methodology and result section of the abstract do not display crucial information such as MIC, SEM analysis, P value. 

Introduction - 

1. I suggest authors break complex sentences into simple sentences to make them more readable. 

2. Mention the mechanism of AgNP acting as an antimicrobial agent. 

Methodology, 

1. Comment on the sample size considered for the study. 

Results

1. Describe the abbreviations used in all graphs and figures in the footnotes. 

2. The legends of figures 1, 5, and 6 are not informative. Authors should give sufficient about the image and its interpretation to the readers, as to what to focus on and its key findings.  

3. Data labels can be added in the graphs for better understanding. 

4. Authors have used Mann Whitney-U test when they are comparing AgNP and control, but they have not mentioned the statistical test used when they were comparing within the study group i.e. AgNP and control group for different types of orthodontic appliances (figure 2b and 2c). 

Discussion

1. Very complex sentences make it difficult to read and comprehend. For example, on Page 11, Line 373-377, a single sentence of four lines.... 

2. The author has mentioned "Several studies", (Line 373) but has not provided any reference. 

Best Wishes

Round 2

Reviewer 1 Report

the manuscript has been correctly revised